# GENRAD: Genomics and Radiomics Heterogeneous Graph Neural Network for Graph-Level Classification in Alzheimer's Disease

## Abstract

Alzheimer's Disease (AD) poses multifaceted challenges due to its neurodegenerative nature driven by complex genomic, radiomic, and structural interactions. Understanding these complex relationships is pivotal for advancing diagnostic and therapeutic approaches. Current models struggle to effectively integrate multimodal data for AD, limiting their predictive accuracy and biological interpretability. Thus, there is a pressing need for models that can seamlessly fuse genomic and radiomic data to provide a holistic understanding of AD pathology. We introduce GENRAD, a novel heterogeneous graph neural network (GNN) that integrates multimodal genomic and radiomic data for graph-level classification in AD by representing patients, genes, and brain structures as distinct nodes and implementing advanced message-passing techniques. The benefits of GENRAD are fourfold: (1) It enables multimodal fusion of genomic and radiomic data, uncovering biologically meaningful insights missed by single-modality models. (2) Its adaptive multi-scale graph representations model interactions at various biological scales, capturing complex relationships essential for understanding AD pathology. (3) GENRAD incorporates explainable AI techniques, providing detailed analysis of key genomic markers and brain regions associated with AD. (4) GENRAD performs unsupervised clustering of genes, allowing the identification of functionally related biological pathways, thus empowering clinicians with actionable insights for personalized treatment strategies. GENRAD demonstrates superior classification accuracy in identifying AD-related patterns compared to existing machine and deep learning models, achieving an accuracy of 91.70%.

## 1 Introduction

Alzheimer's Disease (AD) is a complex neurodegenerative disorder influenced by a combination of genetic, structural, and environmental factors. Early and accurate diagnosis is critical for effective intervention, yet the disease's heterogeneity poses a substantial challenge. Globally, three out of four individuals with dementia remain undiagnosed and untreated. By 2025, the number of people living with dementia is projected to reach 139 million—a threefold increase from current levels Nandi et al. (2022). This rise in cases presents not only a health crisis but also an economic burden, as dementia currently costs the global economy over \$1.3 trillion annually, a figure expected to increase nearly nine-fold by 2025 Pedroza et al. (2022). Given this urgent situation, there is a pressing need to develop early diagnostic methods to delay disease progression and alleviate the strain on healthcare systems. With the public data, models, and code, GENRAD provides an excellent foundation to help advance the analysis of challenging heterogeneous data.

Recent advances in deep learning, particularly Graph Neural Networks (GNNs), offer considerable potential for modeling the convoluted relationships found in biomedical data. However, current GNN models face significant limitations in integrating multimodal data, such as genomic, radiomic, and clinical information. The inherent differences in data types—ranging from genetic sequences to structural MRI scans—pose a significant barrier to their fusion in a unified model. Existing methods often address these modalities separately, leading to suboptimal predictive performance and reduced interpretability Venugopalan et al. (2021); Shi et al. (2022). Moreover, many GNN models struggle

to scale effectively with high-dimensional multimodal data, resulting in computational inefficiency and limited scalability Zhang et al. (2023).

In this paper, we introduce GENRAD, a novel heterogeneous graph neural network designed to seamlessly integrate multimodal genomic and radiomic data for graph-level classification in AD. GENRAD addresses key shortcomings of existing models by representing patients, genes, and brain structures as distinct nodes, thus capturing the underlying biological relationships through sophisticated message-passing techniques. Unlike previous approaches that fuse multimodal data late in the process, GENRAD uses a multi-scale graph representation to capture both local and global interactions across these modalities, providing a comprehensive view of AD pathology.

GENRAD's unique approach offers significant contributions in four critical areas:

1. **Multimodal Fusion for AD:** GENRAD integrates genomic data with structural radiomic features from MRI, significantly boosting predictive accuracy and uncovering biologically meaningful insights that single-modality models would miss.

2. **Adaptive Multi-scale Graph Representations:** By modeling interactions between genes, brain structures, and patients at multiple biological scales, GENRAD captures complex relationships that are essential for understanding AD pathology and disease progression.

3. **Enhanced Explainability and Interpretability:** GENRAD incorporates explainable AI techniques, enabling detailed analysis of key genomic markers and brain regions associated with AD. This empowers clinicians with actionable insights for personalized treatment.

4. **Unsupervised Clustering of Genes:** GENRAD introduces supernodes to perform unsupervised clustering of genes, allowing the identification of functionally related biological pathways and clusters, which improves our understanding of AD-related mechanisms.

Through extensive experiments, we demonstrate that GENRAD outperforms state-of-the-art models while providing interpretable predictions. Our contributions offer a pathway toward early diagnosis and targeted interventions, addressing the growing global burden of AD.

## 2 RELATED WORK

GNNs have gained prominence for their adeptness in handling graph-structured data, proving especially valuable in domains rich with relational data like medical imaging and genomics.

**Graph Neural Networks in Medical Domain.** In medical applications, GNNs excel in capturing complex spatial and relational patterns. For instance, GNNs have been utilized in MRI-based analyses to identify atrophy in specific brain regions, successfully predicting Alzheimer's Disease (AD) from structural MRI data by exploiting the nuanced spatial relationships within the brain Ktena et al. (2017). Prominent GNN architectures adapted for medical applications include Graph Convolutional Networks (GCNs), Graph Attention Networks (GATs), and GraphSAGE. These models have been particularly influential in analyzing brain networks to diagnose neurodegenerative disorders such as AD, Parkinson's Disease, and Autism Spectrum Disorder Parisot et al. (2018). Recent advancements have further demonstrated the utility of GNNs in multimodal neuroimaging and functional MRI data analysis. For example, BrainGNN effectively interprets fMRI data by identifying critical brain regions for AD diagnosis Li et al. (2021), while interpretable temporal graph neural networks leverage longitudinal imaging data for prognostic predictions of AD progression Kim et al. (2021). Similarly, multimodal graph convolutional networks have been employed to combine structural and functional imaging data, offering robust and interpretable tools for clinical diagnosis Zhou et al. (2022); Xiao et al. (2022).

**Multimodal Data Integration for Disease Prediction.** The prediction and diagnosis of complex diseases like AD necessitate the integration of diverse data types, including genetic, radiomic, and clinical data. Multimodal models, designed to capture the multifaceted nature of such diseases, aim to discern patterns unattainable by single-modality approaches. Despite their potential, many existing models struggle to fully leverage the benefits of data integration, often due to challenges in harmonizing spatial information from radiological images with structured genomic data Reymbaut et al. (2021). Recent multimodal GNN approaches, such as those integrating radiomics and genomics, have begun addressing these challenges by harmonizing diverse datasets to improve pre-

dictive accuracy and interpretability. For instance, multicenter and multichannel pooling GCNs successfully merge dual-modality imaging data for early AD diagnosis, while adversarially regularized graph learning models, such as DeepASD, integrate multimodal data for Autism Spectrum Disorder diagnosis Wang et al. (2022); Xiao et al. (2024). Foundational work on anatomical parcellation Tzourio-Mazoyer et al. (2002) and intrinsic functional connectivity mapping Schaefer et al. (2018) supports these models in extracting meaningful spatial features. Benchmarks like NeuroGraph and related data-driven network neuroscience initiatives have further standardized the application of graph machine learning in brain connectomics, promoting reproducibility in multimodal research Said et al. (2023); Xu et al. (2023).

**Explainability in GNNs.** Explainability is paramount in healthcare applications of ML, including the use of GNNs, where understanding model predictions is crucial due to the high stakes of medical decision-making. Explainability techniques such as GNNExplainer isolate critical subgraphs by analyzing node and edge contributions to the classification score and assign importance scores based on their influence on the final prediction Ying et al. (2019). Recent developments have introduced interpretable graph-based methods that elucidate the influence of multimodal data convergence on model predictions. Models such as GENRAD enhance these explainability mechanisms by offering detailed visualizations of how individual modalities contribute to outcomes. The addition of self-attention mechanisms in GNN architectures has further improved the interpretability of disease predictions by focusing on key features in complex brain networks Kazi et al. (2019). Radiogenomics plays a crucial role in these advancements, with applications ranging from cancer diagnostics to neurodegenerative diseases Bodalal et al. (2019); Li & Zhou (2022). Integrative network analyses, such as those identifying molecular signatures underlying regional vulnerabilities in AD, provide valuable insights into selective disease mechanisms Wang et al. (2016).

## 3 METHODOLOGY

In this section, we describe the technical framework of GENRAD, highlighting the key innovations in multimodal data integration and graph-based modeling. GENRAD's design is centered around addressing two primary challenges in AD diagnosis: (1) the effective fusion of heterogeneous data modalities and (2) the capture of complex interactions across biological scales.

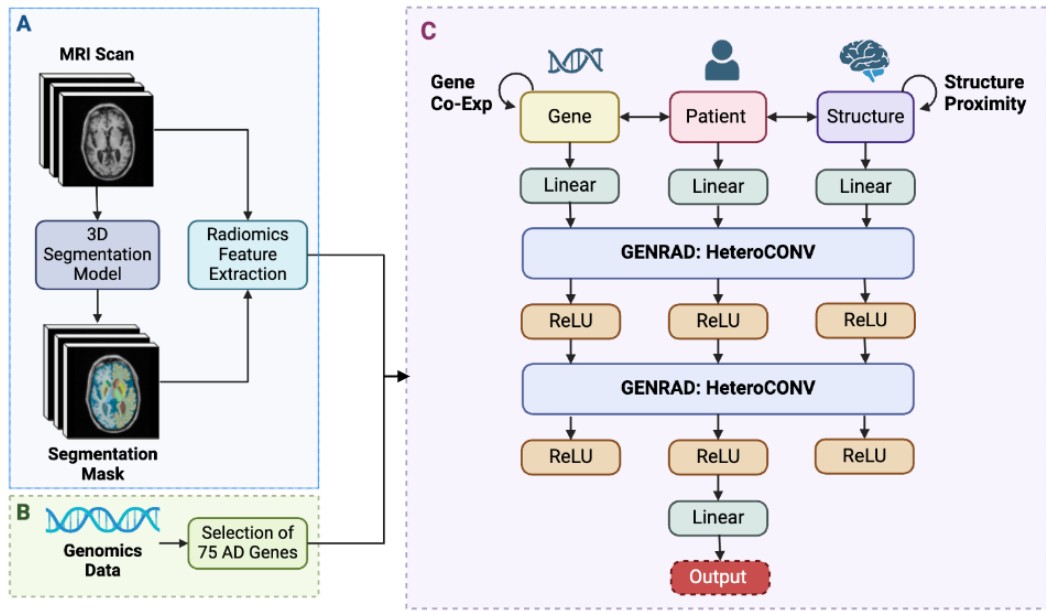

Figure 1: Architecture for heterogeneous GENRAD model. The process involves three main steps: (A) segmenting MRI scans using a 3D UNet model and extracting radiomics features, and (B) selecting relevant genetic data. The combined data feeds into GENRAD (C) featuring gene-to-gene (co-expression), structure-to-structure, and patient-to-structure and patient-to-gene interactions.

## 3.1 Feature Extraction and Pre-processing

Radiomic Features: To derive structural features from brain MRI scans, we first segment the brain into 32 distinct structures using the SynthSeg model, a deep learning tool known for its robustness to variations in MRI resolution and trained using elder and diseased scans Billot et al. (2023). SynthSeg automatically generates segmentation masks, which are validated through a three-step process: (1) automated quality control, (2) comparison of volumes against known ground truths, and (3) manual verification by an expert radiologist. We then use PyRadiomics to extract 107 distinct features from each segmented brain structure, including shape, first-order statistics, and gray-level attributes, providing a detailed radiomic profile for each patient Van Griethuysen et al. (2017).

Genomic Features: We utilize processed genomic data, including expression levels for 75 key genes highly associated with AD, sourced from Human Protein Atlas (2024). The genomic data is scaled and mapped to the appropriate gene names, ensuring consistent representation across patients. The integration of these distinct features enables GENRAD to model both genetic predisposition and structural brain changes, offering a holistic view of AD pathology.

## 3.2 Heterogeneous Graph Structure

GENRAD's core innovation lies in its heterogeneous graph architecture $G = (V, E)$, which simultaneously models patients, genes, and brain structures as distinct node types. This multimodal graph allows for the capture of complex interactions between these entities, including patient-gene and patient-structure associations, structure-to-structure relationships, and gene co-expression patterns. Unlike existing methods, GENRAD adapts to the heterogeneity of these data types through a multi-scale graph structure, which weights interactions at both the local and global levels.

### 3.2.1 Node Representation

**Patients** ($V_p$): Each patient node $p_i \in V_p$ is represented by a feature vector $\mathbf{x}_{p_i} \in \mathbb{R}^{d_p}$, where $d_p$ is the dimension of patient-specific features. The features include the encoded APOE genotype, the gender, the Global Deterioration Scale (GDS), and Mini-Mental State Examination (MMSE) scores. The final feature matrix for patient nodes is $\mathbf{X}_p \in \mathbb{R}^{n_p \times d_p}$, where $n_p$ is the number of patients, captures key clinical indicators that are known to correlate with AD progression.

**Genes** ($V_g$): Each gene node $g_i \in V_g$ is represented by its corresponding expression values across patients. The feature matrix for genes is denoted as $\mathbf{X}_g \in \mathbb{R}^{n_g \times 1}$, where $n_g$ is the number of genes. For each patient, we compute a flattened vector of gene expression values for all associated genes. This feature vector enables GENRAD to capture the molecular aspects of AD by linking patients to relevant genomic information.

**Structures** ($V_s$): Each structure node $s_i \in V_s$ represents a brain region segmented from MRI scans. The radiomic features for each structure are extracted using PyRadiomics, resulting in a feature vector $\mathbf{x}_{s_i} \in \mathbb{R}^{d_s}$, where $d_s$ represents the number of radiomic features (107 features across 32 structures). The final feature matrix for structures is $\mathbf{X}_s \in \mathbb{R}^{n_s \times d_s}$, where $n_s$ is the number of brain structures. This ensures that structural changes in the brain, which are critical in AD, are directly incorporated into the graph model.

### 3.2.2 Edge Construction

The edges between nodes in the graph represent various biological and structural relationships, capturing both local interactions (e.g., patient-gene, patient-structure associations) and global patterns (e.g., gene co-expression and structure-to-structure).

**Patient-Gene Edges** ($E_{pg}$): We model the association between patients and their gene expression data. Each patient $p_i$ is connected to all selected 75 gene nodes $g_j$, forming edges $(p_i, g_j) \in E_{pg}$. The edge index for these edges is stored in the matrix $\mathbf{E}_{pg} \in \mathbb{R}^{2 \times |E_{pg}|}$, where $|E_{pg}|$ is the total number of patient-gene connections. This allows the model to directly incorporate individual genetic variations into the AD prediction task.

**Patient-Structure Edges** ($E_{ps}$): Patient nodes are connected to structure nodes based on the brain regions associated with the patient's MRI data. The relationship is represented by the edge index

matrix $\mathbf{E}_{ps} \in \mathbb{R}^{2 \times |E_{ps}|}$. This enables the model to integrate radiomic features and genetic data, providing a multimodal representation of each patient's disease profile.

**Gene Co-expression Edges** ($E_{gg}$)**:** Gene-gene interactions are modeled through a co-expression network, where edges are established between pairs of genes $(g_i, g_j) \in E_{gg}$ weighted by their co-expression. The edge weight is stored in $E_{gg} \in \mathbb{R}^{2 \times |E_{gg}|}$. The co-expression score was obtained from GeneMANIA Warde-Farley et al. (2010). These weighted edges, based on the strength of gene co-expression, allow the model to account for gene interaction networks that contribute to AD pathology.

**Structure-Structure Edges** ($E_{ss}$)**:** To capture relationships between different brain structures, these connections form the structure-to-structure edges $(s_i, s_j) \in E_{ss}$ with the edge weight matrix $E_{ss} \in \mathbb{R}^{2 \times |E_{ss}|}$. We define edges between each structure and all other 31 structures weighted based on the 3D Euclidean distance between the center of mass of each segmented structure. These edges capture spatial relationships between brain regions that are critical for understanding the progression of AD since there is no functional data available in the dataset.

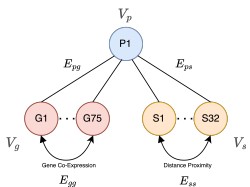
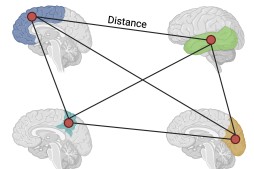

(a) Heterogeneous Graph Structure representing multimodal relationships.

(b) Connection between brain structures according to distance from the center of mass.

Figure 2: Illustration of interpretability using heterogeneous graph and brain structures.

### 3.2.3 GRAPH FORMULATION

The heterogeneous graph $G$ is shown in Figure 2a and is formally represented as:

$$G = \{(V_p, V_g, V_s), (E_{pg}, E_{ps}, E_{gg}, E_{ss})\} \tag{1}$$

where $V_p, V_g, V_s$ represent the patient, gene, and structure node sets, respectively. $E_{pg}, E_{ps}, E_{gg}, E_{ss}$ represent the edges between patients and genes, patients and structures, gene co-expression, and structure-to-structure relationships, respectively.

### 3.3 GENOMIC AND RADIOMIC FEATURE INTEGRATION

A key innovation of GENRAD is its multi-scale integration of genomic and radiomic features. Unlike previous models, which often fuse multimodal data at a late stage, GENRAD incorporates these features into a heterogeneous graph from the start, allowing for direct interaction between gene, structure, and patient nodes. The message-passing mechanism ensures that the information from each modality is propagated through the graph, allowing for adaptive weighting of different data types based on their importance.

### 3.4 MESSAGE PASSING IN GENRAD

Message passing in GENRAD involves propagating information between different node types (patients, genes, and brain structures) in a heterogeneous graph. The goal is to aggregate and update node representations by combining the features of neighboring nodes through learnable aggregation functions. This process is implemented using the SAGEConv layers Hamilton et al. (2017), which are designed to perform efficient neighborhood aggregation by propagating information between heterogeneous nodes. This enables the model to capture complex multimodal relationships in AD pathology by incorporating genomic and radiomic data. The SAGEConv operation ensures that the model can efficiently aggregate and learn from high-dimensional multimodal data.

### 3.4.1 Aggregating Features Across Node Types

In the heterogeneous graph, we define three node types: patients ($V_p$), genes ($V_g$), and brain structures ($V_s$). Edges connect these nodes and message-passing aggregates information from the neighbors of each node type. For example, patient nodes aggregate information from their connected gene and structure nodes, ensuring that both molecular and structural features are integrated into the final patient representation. This aggregation process is crucial for capturing the synergistic effects of radiomic and genomic features, enabling the model to discover deeper insights into AD pathology.

### 3.4.2 SAGEConv Layer for Node Updates

We utilize the SAGEConv layer to perform the aggregation and update operations. For each node, the SAGEConv layer aggregates information from its neighbors and updates the node's feature vector based on this aggregated information. The general update rule for node:

$$\mathbf{h}_i^{(t+1)} = \sigma \left( \mathbf{W}\mathbf{h}_i^{(t)} + \text{AGGREGATE}\left( \{\mathbf{h}_j^{(t)} : j \in \mathcal{N}(i)\} \right) \right) \quad (2)$$

Where $\mathbf{h}_i^{(t)}$ is the feature vector of node $i$ at time step $t$, $\mathbf{W}$ is the learnable weight matrix, $\sigma$ is a non-linear activation function (e.g., ReLU), and AGGREGATE represents an aggregation function (e.g., mean or sum) applied to the neighboring node features. This allows the model to adaptively aggregate information from neighboring nodes, making it capable of handling high-dimensional multimodal data efficiently.

### 3.4.3 Message Passing Mathematical Formulation

The message-passing process involves aggregating information from neighboring nodes. The generalized message-passing formula can be applied to any node type in the heterogeneous graph and easily extended to new modalities. For any node $v_i$ of type $\tau(i)$, the message passing update rule can be expressed as:

$$\mathbf{h}_i^{(t+1)} = \sigma \left( \mathbf{W}_{\tau(i)} \mathbf{h}_i^{(t)} + \sum_{\phi \in \Phi} \sum_{j \in \mathcal{N}\phi(i)} \mathbf{W}_{\tau(i),\phi} \mathbf{h}_j^{(t)} \right) \quad (3)$$

Where $\tau(i)$ is the type of node $i$ (e.g., patient, gene, structure, or any additional modality), $\Phi$ is the set of all neighboring node types in the graph, $\mathcal{N}_\phi(i)$ represents the set of neighboring nodes of type $\phi$ for node $i$, $\mathbf{W}_{\tau(i)}$ is the self-connection weight matrix for nodes of type $\tau(i)$. $\mathbf{W}_{\tau(i),\phi}$ is the weight matrix for messages from nodes of type $\phi$ to nodes of type $\tau(i)$, $\sigma$ is a non-linear activation function. In this setup, $\Phi_{patient} = \{patient, gene, structure\}$, $\Phi_{gene} = \{patient, gene\}$, and $\Phi_{structure} = \{patient, structure\}$.

### 3.5 Graph-Level Classification

The GENRAD model performs graph-level classification by leveraging the integrated feature representations of patient nodes, where each patient node represents the entire graph's multimodal information. After multiple rounds of message passing between patients, genes, and brain structures, the final patient node encapsulates the aggregated data from the entire graph, including genomic, radiomic, and structural relationships, which is then used to predict the patient's AD status.

**Patient Representation Aggregation.** For each patient node $p_i$, the final feature representation $\mathbf{h}_i^{(t)}$ is obtained after aggregating information from its neighboring gene nodes $g_j$ and structure nodes $s_k$. This aggregation occurs through multiple message-passing iterations, where each patient's representation is updated by combining features from both genomic and radiomic contexts. The final patient feature vector is computed as illustrated in Equation 3.

**Classification Layer.** Once the patient node embedding $\mathbf{h}_i^{(t)}$ is obtained, it is passed through a fully connected classification layer to predict the patient's AD status. The softmax function is applied to produce a probability distribution over the possible disease class: $\hat{y}_i =$

softmax $\left(\mathbf{W}_c\mathbf{h}_i^{(t)} + \mathbf{b}_c\right)$, where $\hat{y}_i$ is the predicted probability distribution for the disease class of patient $p_i$, $\mathbf{W}_c$ is the learnable weight matrix of the classification layer, and $\mathbf{b}_c$ is the bias term. The model outputs the probability for each disease class, and the class with the highest probability is selected as the label.

**Contribution of Genomic and Radiomic Contexts.** The patient node representation $\mathbf{h}_i^{(t)}$ is a comprehensive feature vector that captures both genomic and radiomic information. Gene expression data, encoded by the gene nodes, contributes critical insights into the molecular factors influencing AD. At the same time, radiomic features extracted from brain MRI scans provide structural information about the brain. By integrating both data types, GENRAD leverages the full scope of multimodal information to make more accurate and meaningful predictions about AD progression.

**Loss Function.** The model is trained using cross-entropy loss, which measures the difference between the predicted probability distribution and the true label distribution. The loss for each patient is defined as: $\mathcal{L} = -\sum_{c=1}^{C} y_{i,c} \log(\hat{y}_{i,c})$, where $C$ is the number of classes, $y_{i,c}$ is the true label for patient $p_i$ for class $c$, and $\hat{y}_{i,c}$ is the predicted probability for class $c$.

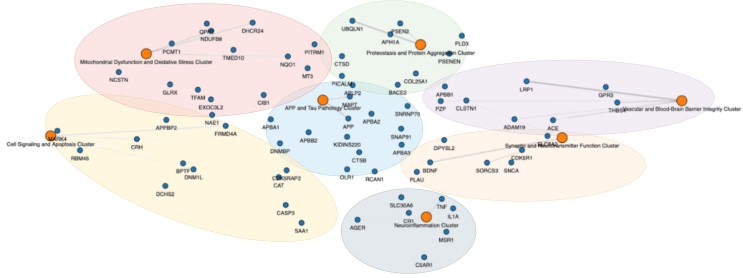

Figure 3: Network visualization depicting gene clustering based on their functional associations within AD pathways. The supernodes (orange) represent distinct biological clusters. Genes (blue nodes) are connected within and across clusters.

# 4 EXPLAINABILITY IN GENRAD

GENRAD incorporates explainable AI techniques alongside biological interpretability frameworks to ensure its predictions are not only accurate but also clinically meaningful.

**GNNExplainer.** The key innovation here is GENRAD's ability to isolate critical sub-graphs and features that contribute to the prediction of AD using GNNExplainer. This tool is crucial for identifying important gene-gene interactions, patient-gene relationships, and connections between structural brain regions to clarify the underlying biological mechanisms driving the model's predictions.

**Biological Interpretability and Clinical Utility.** GENRAD offers both technical explainability and biological interpretability, uncovering potential new biomarkers and clarifying known mechanisms related to AD. By highlighting crucial gene pathways, such as those involved in amyloid-beta processing (e.g., APP, PSEN1), GENRAD links genetic activities to observable structural changes in the brain, particularly in key regions like the hippocampus. Additionally, GENRAD's explainability supports clinical decision-making by pinpointing critical biomarkers such as APOE.

**Synergy Between Explainability and Multimodal Fusion.** GENRAD's ability to articulate its predictions benefits significantly from its integration of genomic and radiomic data, enhancing both accuracy and biological insight. Figure 4 enhances GENRAD's explainability, demonstrating its capability to visually integrate and interpret complex genetic and structural data interactions, thereby making its scientific insights actionable in clinical settings.

**Unsupervised Gene Supernode Clustering.** This higher level of abstraction allows for the grouping of genes based on shared co-expression patterns and functions. As illustrated by the chord Figure 5, genes are grouped into distinct biological clusters, including APP and Tau pathology and neuroinflammation, among others. The analysis highlighted significant cross-talk between pathways like mitochondrial dysfunction and cell signaling, which suggests their co-involvement in neurodegenerative processes. Notably, clusters such as vascular integrity and protein aggregation display

strong connections to APP, MAPT, and related genes, reflecting their central role in AD. Further information is provided in the Appendix and illustrated in Figure 3.

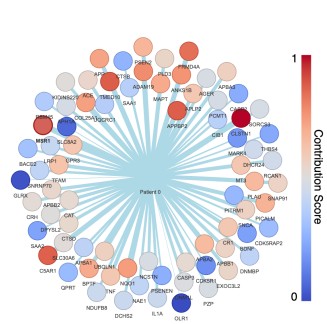
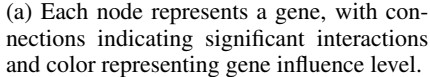
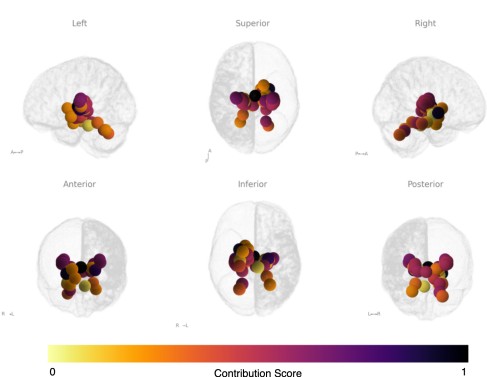

(a) Each node represents a gene, with connections indicating significant interactions and color representing gene influence level.

(b) Each sphere represents the center of structures, and color represents the level of influence.

Figure 4: Interpretability of Graph Models in Multimodal Analysis (a) Network visualization of gene interactions highlighting influential genes in AD pathogenesis. (b) Corresponding radiomics-based analysis of brain regions affected in AD, depicted through MRI scans with overlaid heatmaps.

## 5 EXPERIMENTS

### 5.1 DATASET

Building a large-scale dataset targeting dementia sub-types is expensive and time-consuming. There is only one public dataset to our knowledge that offers a rich foundation for dementia research, which is ANMerge Birkenbihl et al. (2021); Table 7 in the appendix outlines the dataset's key statistics. Unlike the ADNI dataset, it encompasses longitudinal MRI scans and comprehensive genomic data. There are four classes: AD, Vascular Dementia (VaD), Mild Cognitive Impairment (MCI), and Control (CTL). Given its recent publication, there are limited papers available on ANMerge, and it only contains one neuroimaging modality, which is structural MRI, and no functional information.

### 5.2 EXPERIMENTAL SETUP

The GENRAD model is trained using a stratified 3-fold cross-validation approach to ensure a balanced representation of the classes in each fold. For training, the Adam optimizer with a learning rate of 0.001 and a weight decay of $5 * 10^{-4}$ was used. Early stopping is employed with a patience of 25 epochs to avoid overfitting. The multimodal dataset is built using PyTorch Geometric's Hetero-Data class, Fey & Lenssen (2019). The class imbalance was handled using class weights computed for the cross-entropy loss function. The hidden channel size for the SAGEConv layers was set to 64.

### 5.3 ABLATION STUDY

The ablation study first evaluated the baseline performance of a 3D CNN model using raw MRI, establishing a benchmark for the effectiveness of image-based features in isolation. We then extended our analysis by testing various ML and DL models (Random Forest, Support Vector Machine, and Feature Generation by Convolutional Neural Network) to gauge their performance relative to GENRAD; the results are provided in the appendix. Our ablation study rigorously evaluated GENRAD's components, focusing on multimodal integration, graph edges, and message-passing. We showed that combining genomic and radiomic data improves classification accuracy compared to using them independently. Analysis of graph edges (patient-gene, patient-structure, gene co-expression, structure-structure) revealed that removing any edge type reduced performance, underscoring the importance of capturing both local and global biological relationships. The SAGEConv message-passing layers were critical, as detailed in Appendix Table 9.

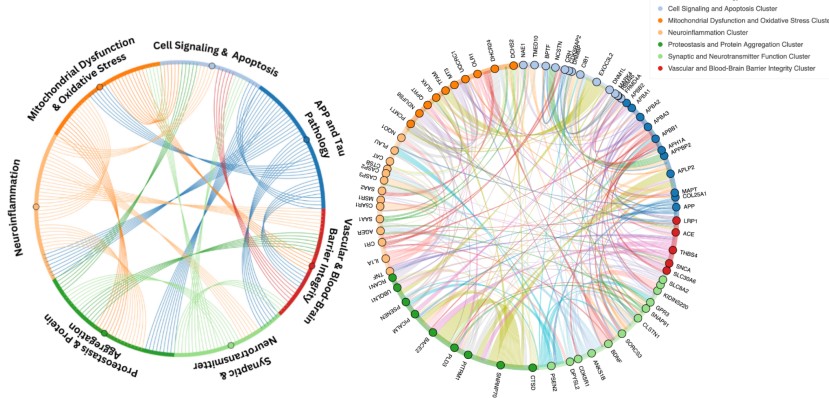

Figure 5: Left: A chord diagram visualizing the high-level relationships between supernodes, representing distinct biological clusters. The connections indicate the degree of interplay between them. Right: Gene-level connections highlighting detailed interactions among genes.

# 6 RESULTS AND DISCUSSION

| Model | Dataset | Metric | AD Vs CTL | AD Vs MCI | MCI Vs CTL | AD Vs VaD |
|---|---|---|---|---|---|---|
| **Maddalena et al. (2023)** | **MRI + Gen** | Accuracy | 94.10 | 91.80 | 61.70 | - |
| | | F1-Score | 76.40 | 48.70 | 71.20 | - |
| | | Recall | 68.80 | 56.00 | 67.70 | - |
| | | Precision | 88.30 | 45.50 | 75.30 | - |
| **3D CNN** | **MRI** | Accuracy | 84.27 ± 5.32 | 72.01 ± 5.68 | 77.73 ± 4.52 | 72.83 ± 5.62 |
| | | F1-Score | 80.50 ± 4.85 | 70.96 ± 5.12 | 77.89 ± 3.98 | 69.91 ± 4.51 |
| | | Recall | 79.67 ± 5.02 | 70.08 ± 4.21 | 77.77 ± 4.09 | 69.56 ± 3.84 |
| | | Precision | 81.34 ± 4.85 | 71.87 ± 5.20 | 78.02 ± 4.65 | 70.26 ± 4.76 |
| **SVM** | **Rad + Gen** | Accuracy | 80.46 ± 3.70 | 61.47 ± 3.90 | 69.90 ± 3.95 | 81.71 ± 4.29 |
| | | F1-Score | 80.89 ± 3.10 | 52.55 ± 3.85 | 69.85 ± 3.81 | 85.31 ± 5.26 |
| | | Recall | 83.72 ± 4.30 | 52.99 ± 7.24 | 69.89 ± 5.21 | **91.72 ± 4.32** |
| | | Precision | 78.26 ± 3.70 | 53.11 ± 3.36 | 69.83 ± 3.75 | 79.76 ± 3.85 |
| **FGCNN** | **Rad + Gen** | Accuracy | 97.92 ± 2.15 | 90.69 ± 1.65 | 86.45 ± 2.87 | **89.25 ± 5.62** |
| | | F1-Score | 97.92 ± 3.06 | 90.68 ± 1.87 | 84.23 ± 3.84 | 81.13 ± 4.58 |
| | | Recall | 97.91 ± 2.54 | 90.69 ± 1.85 | 82.63 ± 4.02 | 83.61 ± 3.89 |
| | | Precision | 98.92 ± 3.20 | 90.85 ± 1.87 | **87.24 ± 4.11** | 79.21 ± 3.88 |
| **GENRAD (Ours)** | **Rad + Gen** | Accuracy | **98.87 ± 1.60** | **93.42 ± 1.75** | **89.63 ± 2.92** | 88.25 ± 1.96 |
| | | F1-Score | **98.91 ± 1.59** | **92.80 ± 1.80** | **86.93 ± 2.21** | **87.84 ± 2.24** |
| | | Recall | **98.87 ± 1.48** | **92.35 ± 1.82** | **88.60 ± 1.98** | 86.32 ± 2.11 |
| | | Precision | **98.95 ± 1.60** | **93.25 ± 1.79** | 85.32 ± 2.05 | **89.42 ± 1.85** |

Table 1: Performance comparison of different models across various metrics and classification tasks across ANMerge datasets with varying data types. Hyphens indicate unreported metrics. The standard deviation for Maddalena et al. (2023) was not reported.

The results from the ablations in Table 6 and from GENRAD in Table 2 underscore significant advancements in the integration of multimodal data for the classification of AD.

**Comparison.** GENRAD outperforms traditional models in classifying AD by leveraging multimodal genomic and radiomic data. Unlike the 3D CNN, which focuses solely on MRI, or the SVM, which integrates radiomics and genomics but lacks synergy, GENRAD achieves higher accuracy. GENRAD also outperforms FGCNN, demonstrating superior results across all metrics with a notable accuracy of 98.87%, compared to the 84.27%, 80.46%, and 97.92% achieved by the 3D CNN, SVM, and FGCNN, respectively. The previous work by Maddalena et al. (2023) achieved competitive results using a combination of MRI and genomic data, with an accuracy of 94.1% for AD vs CTL. However, compared to GENRAD, which demonstrates superior performance across a broader range of metrics and tasks, Maddalena's model falls short in handling more complex classifications like MCI vs CTL, whereas GENRAD's multimodal integration offers better results. Although it is

| Data | | Edge Connections | | Classes | | | | Metrics | | | |
|---|---|---|---|---|---|---|---|---|---|---|---|
| Gen | Rad | Struct | Co-Exp | CTL | MCI | AD | VaD | Accuracy (%) | F1-Score (%) | Precision (%) | Recall (%) |
| ○ | ● | ○ | ○ | ● | ○ | ● | ○ | 66.67 ± 4.23 | 64.57 ± 4.15 | 68.88 ± 5.93 | 66.67 ± 4.23 |
| ○ | ● | ● | ○ | ● | ○ | ● | ○ | 77.23 ± 2.11 | 78.26 ± 1.71 | 79.32 ± 3.93 | 77.23 ± 2.11 |
| ○ | ● | ○ | ○ | ● | ● | ● | ● | 55.37 ± 2.11 | 43.40 ± 3.32 | 37.93 ± 8.08 | 55.37 ± 2.11 |
| ○ | ● | ● | ○ | ● | ● | ● | ● | 57.06 ± 5.24 | 52.43 ± 5.62 | 50.28 ± 4.43 | 57.06 ± 5.24 |
| ● | ○ | ○ | ○ | ● | ○ | ● | ○ | 85.31 ± 3.48 | 85.21 ± 3.63 | 87.15 ± 3.77 | 85.31 ± 3.48 |
| ● | ○ | ○ | ● | ● | ○ | ● | ○ | 89.27 ± 3.62 | 89.22 ± 3.66 | 91.64 ± 2.92 | 89.27 ± 3.62 |
| ● | ○ | ○ | ○ | ● | ● | ● | ● | 80.33 ± 2.98 | 81.11 ± 2.69 | 81.90 ± 1.91 | 80.33 ± 2.98 |
| ● | ○ | ○ | ● | ● | ● | ● | ● | 82.60 ± 2.20 | 81.09 ± 3.29 | 80.49 ± 4.42 | 82.60 ± 2.20 |
| ● | ● | ○ | ● | ● | ○ | ● | ○ | 91.14 ± 4.79 | 90.73 ± 4.81 | 90.33 ± 5.03 | 91.14 ± 4.79 |
| ● | ● | ● | ○ | ● | ○ | ● | ○ | 93.79 ± 1.80 | 93.79 ± 1.81 | 93.96 ± 2.05 | 93.79 ± 1.80 |
| ● | ● | ● | ● | ● | ○ | ● | ○ | **98.87 ± 1.60** | **98.91 ± 1.59** | **98.95 ± 1.48** | **98.87 ± 1.60** |
| ● | ● | ○ | ● | ● | ● | ● | ● | 85.37 ± 2.11 | 84.90 ± 0.82 | 84.34 ± 4.44 | 85.37 ± 2.11 |
| ● | ● | ● | ○ | ● | ● | ● | ● | 84.50 ± 1.60 | 81.37 ± 1.74 | 78.67 ± 1.90 | 84.50 ± 1.60 |
| ● | ● | ● | ● | ● | ● | ● | ● | **91.79 ± 3.62** | **91.70 ± 3.52** | **92.02 ± 3.52** | **91.79 ± 3.62** |

Table 2: Breakdown of GENRAD results to evaluate the impact of data modalities, edge connections, classification tasks, and corresponding performance metrics. The filled circle indicates the inclusion in the analysis, and the empty circle indicates the exclusion.

a different dataset, due to the limited published comparisons using ANMerge, we can also compare GENRAD's performance to ADNI-based models. GENRAD outperforms traditional models, which only achieved 91.30% by Zheng et al. (2018) and 94.60% by Maddalena et al. (2022) on the ADNI dataset. Thus, confirming that GENRAD's multimodal approach yields superior results.

**Multimodal Synergy.** The significant improvement in performance when integrating radiomics and genomics data compared to individual modalities highlights the strong synergistic effect of multimodal data fusion. This demonstrates GENRAD's capacity to extract complementary information, where radiomics captures structural brain changes, and genomics reveals genetic factors associated with AD. By harmonizing these distinct data types, GENRAD is able to provide a holistic understanding of AD progression. GENRAD's integration is not just additive but rather synergistic, capturing deeper and more nuanced interactions between genetic and structural features.

**Graph Structure Contribution.** The additional performance gains achieved through the inclusion of structure-to-structure and gene co-expression connections emphasize the value of modeling nuanced relationships in biological data. The graph-based structure allows GENRAD to uncover higher-order interactions, which are often missed by traditional machine learning models. These connections enable the model to capture how different brain regions interact, as well as how genes co-express and contribute to disease progression. The incorporation of these relationships transforms GENRAD into a powerful tool capable of reflecting the complex biological systems underlying AD rather than relying on isolated feature analysis.

**Interpretability and Biological Insights.** The qualitative results presented in Figure 4 and 5 showcase GENRAD's ability to provide biologically meaningful insights. The visualization of gene interactions and affected brain regions aligns with known AD pathology, potentially identifying novel biomarkers or therapeutic targets. GENRAD's high performance makes it a promising tool for early AD diagnosis and personalized treatment planning.

## 7 CONCLUSION

The GENRAD model represents a significant advancement in multimodal integration for AD classification. By leveraging heterogeneous GNN to combine genomic and radiomic data, GENRAD achieves high accuracy in distinguishing between different stages of cognitive decline. The model's performance demonstrates the power of graph-based approaches in capturing complex biological relationships, outperforming individual modalities, and showcasing the synergistic effect of multimodal data integration. The GENRAD model's interpretability is one of its standout features, offering clear insights into gene-gene interactions and the brain regions most impacted by AD, enhancing its potential for clinical application, and furthering our understanding of AD pathogenesis. While promising, future work should focus on validating GENRAD on larger, more diverse datasets incorporating longitudinal data to capture disease progression and integrating other omics data types, such as proteomics, to further enhance the model's predictive power. Finally, a comparative analysis with transformer-based models could be conducted to test their performance against GNNs.

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

# A APPENDIX

## A.1 MACHINE LEARNING RESULTS

| Data | Metric | AD Vs CTL | | AD Vs MCI | | MCI Vs CTL | | AD Vs VaD | |
|---|---|---|---|---|---|---|---|---|---|
| | | RF | SVM | RF | SVM | RF | SVM | RF | SVM |
| **MRI** | Accuracy | 66.69 | 70.57 | 57.77 | 59.12 | 57.95 | 52.38 | 67.55 | 78.20 |
| | F1-Score | 66.99 | 75.86 | 59.45 | 61.89 | 54.36 | 49.04 | 53.67 | 62.17 |
| | Recall | 65.48 | 74.88 | 60.86 | 60.43 | 53.38 | 40.84 | 62.01 | 53.42 |
| | Precision | 68.69 | 76.87 | 58.56 | 63.44 | 60.27 | 61.37 | 47.47 | 71.57 |
| | AUC | 66.65 | 68.36 | 57.58 | 58.98 | 57.59 | 53.98 | 65.98 | 71.11 |
| **Rad+Gen** | Accuracy | 72.51 | 80.46 | 59.18 | 61.47 | 60.65 | 69.90 | 70.25 | 81.71 |
| | F1-Score | 65.35 | 80.89 | 59.18 | 52.55 | 64.19 | 69.89 | 70.25 | 85.31 |
| | Recall | 65.18 | 83.72 | 59.19 | 52.99 | 63.52 | 69.89 | 70.27 | 91.72 |
| | Precision | 66.58 | 78.26 | 59.17 | 53.11 | 64.87 | 69.93 | 70.29 | 79.76 |
| | AUC | 71.67 | 85.77 | 63.76 | 59.76 | 70.17 | 72.65 | 77.84 | 83.47 |

Table 3: Machine Learning Algorithm Results - Summary of result using only MRI scans versus with fusion with multi-omics data for support vector machine (SVM) and random forest (RF) classifiers.

## A.2 IMAGE-BASED BASELINE

| Data | Metric | AD Vs CTL | AD Vs MCI | MCI Vs CTL | AD Vs VaD |
|---|---|---|---|---|---|
| **MRI** | Accuracy | 84.27 | 72.01 | 77.73 | 72.83 |
| | F1-Score | 80.50 | 70.96 | 77.89 | 69.91 |
| | Recall | 79.67 | 70.08 | 77.77 | 69.56 |
| | Precision | 81.34 | 71.87 | 78.02 | 70.26 |
| | AUC | 94.17 | 81.05 | 82.54 | 82.36 |

Table 4: Direct Image Based 3D CNN Results - Table summarizing the accuracy, f1-score, recall, precision, and AUC result of using 3D CNN. The results showcase this limitation compared to the proposed MINDSETS approach.

The state-of-the-art 3D CNN with DenseNet121 backbone achieved an accuracy of 72.83% for the classification of AD vs. VaD and an accuracy of 84.27% for differentiating between AD and CTL, as shown in Table 4.

## A.3 Multi-omics Deep Feature Generation Baseline

| Data | Metric | AD Vs CTL | | AD Vs MCI | | MCI Vs CTL | | AD Vs VaD | | All 4 Classes | |
|---|---|---|---|---|---|---|---|---|---|---|---|
| | | All | MRI at time 0 | All | MRI at time 0 | All | MRI at time 0 | All | MRI at time 0 | All | MRI at time 0 |
| **MRI** | Accuracy | 97.89 | 98.64 | 82.86 | 87.60 | 83.01 | 83.32 | 82.52 | 87.60 | 52.39 | 62.43 |
| | F1-Score | 97.86 | 98.59 | 82.44 | 87.41 | 81.78 | 81.84 | 72.36 | 79.03 | 48.32 | 54.58 |
| | Recall | 98.26 | 98.89 | 82.14 | 88.83 | 80.93 | 83.28 | 76.29 | 82.64 | 58.40 | 61.38 |
| | Precision | 97.65 | 98.33 | 83.20 | 87.25 | 81.95 | 81.04 | 70.35 | 76.61 | 58.55 | 67.64 |
| | AUC | 98.16 | 98.89 | 85.30 | 88.83 | 83.93 | 83.24 | 76.29 | 82.64 | 61.78 | 70.04 |
| **Rad+Gen** | Accuracy | 99.35 | 97.92 | 89.52 | **90.69** | 83.89 | 86.45 | 88.60 | **89.25** | 70.12 | **72.23** |
| | F1-Score | 99.31 | 97.92 | 89.27 | **90.68** | 81.88 | 84.23 | 63.14 | **81.13** | 70.19 | **68.29** |
| | Recall | 99.33 | 97.91 | 90.24 | 90.69 | 81.24 | 82.63 | 59.02 | 83.61 | 75.25 | 66.77 |
| | Precision | 99.29 | 98.92 | 88.89 | 90.85 | 82.75 | 87.24 | 70.92 | 79.21 | 67.96 | 75.62 |
| | AUC | 99.33 | 99.91 | 90.69 | 90.24 | 81.24 | 82.63 | 77.07 | 83.61 | 71.26 | 77.32 |

Table 5: Summary of result using all longitudinal MRI scans, and scan 0 alone versus using it in with other multi-omics data with the DFG module. Results are shown for the binary groups tested and the multiclass experiment.

This table presents a comparison of results between two types of data inputs, MRI radiomics and multi-omics data for scans at different time points. The results are shown for both binary classification (AD vs CTL, AD vs MCI, etc.) and a multi-class setting (All 4 Classes). All 107 radiomics features were used, and for the genomics features, the top genes after feature selection were used and concatenated with the radiomics features to represent the multi-omic approach. For each classification task, performance metrics such as Accuracy, F1-Score, Recall, Precision, and AUC (Area Under the Curve) are reported. The metrics are shown for two conditions: using all available MRI scans versus MRI scans at time 0, and using multi-omics data.

## A.4 Supernode and Gene Connections

Supernodes were introduced to represent higher-order groupings of genes based on known biological functions, allowing for the categorization of genes into meaningful clusters. These supernodes capture major Alzheimer's disease-related processes, such as neuroinflammation, mitochondrial dysfunction, and APP and Tau pathology. By incorporating supernodes into the graph, the GENRAD GNN model identified these clusters to group genes based on their co-expression and functional roles within these pathways. This allows for more interpretable insights into the complex network of gene interactions.

**Significant Insights**

- Neuroinflammation Cluster: Genes such as CASP2, SAA1, and C5AR1 demonstrate strong intra-cluster connectivity, highlighting the role of immune responses in Alzheimer's pathology.

- APP and Tau Pathology Cluster: Genes like APP, MAPT, and APBA1 are central to the Alzheimer's hallmark proteins, reflecting their pivotal role in disease progression.

- Vascular and Blood-Brain Barrier Integrity Cluster: With genes like ACE, LRP1, and NCSTN, this cluster underscores the critical role of vascular health and blood-brain barrier integrity in Alzheimer's. The impairment of these pathways can exacerbate neurodegenerative processes by allowing harmful substances to enter the brain, promoting inflammation and cellular damage.

- Synaptic and Neurotransmitter Function Cluster: Genes like SLC30A6 and GPR3 are essential for neurotransmitter signaling and synaptic integrity. This cluster indicates how disruptions in synaptic function may contribute to cognitive decline in Alzheimer's Disease. The presence of SNCA (alpha-synuclein), a gene also implicated in Parkinson's disease, highlights potential overlapping mechanisms in neurodegenerative disorders.

- Proteostasis and Protein Aggregation Cluster: The cluster, anchored by genes such as PSEN2 (Presenilin-2) and UBQLN1 (Ubiquilin-1), illustrates the importance of protein homeostasis. Dysregulated proteostasis and protein aggregation are central to Alzheimer's pathology, particularly in the formation of amyloid plaques and neurofibrillary tangles.

- Mitochondrial Dysfunction and Oxidative Stress Cluster: Genes like NDUFB8 and UQCRC1 within this cluster emphasize the role of mitochondrial function and oxidative stress in neuronal degeneration. Mitochondrial dysfunction is a critical factor in the energy depletion and cell death seen in Alzheimer's patients.

- Cell Signaling and Apoptosis Cluster: Genes such as CDK5RAP2 and MARK4 are crucial for cell cycle regulation and apoptosis. Aberrant cell signaling and programmed cell death contribute to neuronal loss in Alzheimer's Disease, making this a key cluster of interest.

## A.5 ADNI DATASET REFERENCE

| Model | Dataset | Data | Metric | AD Vs CTL | AD Vs MCI | MCI Vs CTL | AD Vs VaD |
|---|---|---|---|---|---|---|---|
| **Zheng et al. (2018)** | **ADNI** | **MRI** | Accuracy | 91.30 | 73.80 | 97.9 | - |
| | | | F1-Score | - | - | - | - |
| | | | Recall | 93.4 | 64.10 | 98.6 | - |
| | | | Precision | - | - | - | - |
| **Maddalena et al. (2022)** | **ADNI** | **MRI + Gen** | Accuracy | 94.60 | 91.50 | 63.60 | - |
| | | | F1-Score | 78.70 | 44.80 | 70.40 | - |
| | | | Recall | 72.20 | 39.40 | 73.20 | - |
| | | | Precision | 88.90 | 54.60 | 70.40 | - |

Table 6: Performance comparison of different models across various metrics and classification tasks across ADNI datasets with varying data types. Hyphens indicate unreported metrics.

## A.6 ANMERGE DATASET DESCRIPTION

| Features | AD (132) | MCI (93) | CN (165) | VaD (75) | Total (465) |
|---|---|---|---|---|---|
| **Sex** | | | | | |
| Male # (%) | 67 (50.77) | 29 (31.52) | 48 (29.03) | 21 (28.00) | 165 (35.48) |
| Female # (%) | 65 (49.23) | 64 (68.48) | 117 (70.97) | 54 (72.00) | 300 (64.52) |
| **Age (years)** | | | | | |
| Mean (std) | 75.45 (6.60) | 73.96 (5.74) | 72.54 (6.64) | 71.86 (6.32) | 73.98 (6.4) |
| Range | [58,88] | [56,86] | [52,87] | [52,88] | [52,88] |
| **Education (years)** | | | | | |
| Mean (std) | 7.99 (3.98) | 8.97 (4.29) | 11.01 (4.88) | 10.85 (5.61) | 9.32 (4.56) |
| Range | [2,22] | [0,20] | [2,25] | [2,25] | [0,25] |
| **MMSE** | | | | | |
| Mean (std) | 20.80 (4.67) | 27.09 (1.72) | 29.07 (1.2) | 22.60 (5.28) | 25.71 (4.54) |
| Range | [12,30] | [24,30] | [25,30] | [14,30] | [12,30] |

Table 7: Demographic and clinical features by diagnosis group.

### A.7 COMPUTATIONAL EFFICIENCY RESULTS

| Metric | GENRAD (GraphSAGE) | GCN |
|---|---|---|
| *Graph Structure* | | |
| Total Nodes ($|V|$) | 572 | 572 |
| Total Edges ($|E|$) | 102,781 | 102,781 |
| Input Feature Dimension ($d_{in}$) | 64 | 64 |
| Number of Classes (C) | 4 | 4 |
| *Computational Complexity* | | |
| FLOPs per Layer | 31.68M | 39.84M |
| Total FLOPs (3 layers) | 95.04M | 119.52M |
| Parameters | 106.7K | 148.3K |
| *Runtime Performance* | | |
| Inference Time (ms/sample) | 3.2 | 4.8 |
| Memory Usage (GB) | 1.4 | 1.8 |

Table 8: Computational comparison between GENRAD and GCN. GENRAD achieves better efficiency through optimized message passing and heterogeneous graph structure. FLOPs calculated for both models include message passing and node-wise operations across three layers. Inference time measured on NVIDIA GeForce RTX 4090 GPU with batch size 32.

### A.8 COMPARATIVE RESULTS FOR MESSAGE PASSING TECHNIQUES

| Data | | Edge Connections | | Classes | | | | Model | Metrics | | | |
|---|---|---|---|---|---|---|---|---|---|---|---|---|
| Gen | Rad | Struct | Co-Exp | CTL | MCI | AD | VaD | GNN | Accuracy (%) | F1-Score (%) | Precision (%) | Recall (%) |
| ○ | ● | ○ | ○ | ● | ○ | ● | ○ | SAGE | **66.67 ± 4.23** | **64.57 ± 4.15** | **68.88 ± 5.93** | **66.67 ± 4.23** |
| | | | | | | | | GAT | 63.45 ± 4.89 | 61.32 ± 6.12 | 65.54 ± 6.12 | 63.45 ± 4.89 |
| | | | | | | | | GCN | 61.23 ± 4.56 | 59.18 ± 4.44 | 63.42 ± 5.87 | 61.23 ± 4.56 |
| ○ | ● | ○ | ○ | ● | ● | ● | ● | SAGE | **55.37 ± 2.11** | **43.40 ± 3.32** | **37.93 ± 8.08** | **55.37 ± 2.11** |
| | | | | | | | | GAT | 52.15 ± 2.43 | 40.18 ± 3.65 | 34.71 ± 8.42 | 52.15 ± 2.43 |
| | | | | | | | | GCN | 50.04 ± 2.67 | 38.06 ± 3.89 | 32.59 ± 8.76 | 50.04 ± 2.67 |
| ○ | ● | ● | ○ | ● | ○ | ● | ○ | SAGE | **77.23 ± 2.11** | **78.26 ± 1.71** | **79.32 ± 3.93** | **77.23 ± 2.11** |
| | | | | | | | | GAT | 74.01 ± 2.43 | 75.04 ± 2.04 | 76.10 ± 4.27 | 74.01 ± 2.43 |
| | | | | | | | | GCN | 71.90 ± 2.67 | 72.93 ± 2.28 | 73.99 ± 4.51 | 71.90 ± 2.67 |
| ○ | ● | ● | ○ | ● | ● | ● | ● | SAGE | **57.06 ± 5.24** | **52.43 ± 5.62** | **50.28 ± 4.43** | **57.06 ± 5.24** |
| | | | | | | | | GAT | 53.84 ± 5.56 | 49.21 ± 5.95 | 47.06 ± 4.77 | 53.84 ± 5.56 |
| | | | | | | | | GCN | 51.73 ± 5.80 | 47.10 ± 6.19 | 44.95 ± 5.01 | 51.73 ± 5.80 |
| ● | ○ | ○ | ○ | ● | ○ | ● | ○ | SAGE | **85.31 ± 3.48** | **85.21 ± 3.63** | **87.15 ± 3.77** | **85.31 ± 3.48** |
| | | | | | | | | GAT | 82.09 ± 3.80 | 81.99 ± 3.96 | 83.93 ± 4.11 | 82.09 ± 3.80 |
| | | | | | | | | GCN | 79.98 ± 4.04 | 79.88 ± 4.20 | 81.82 ± 4.35 | 79.98 ± 4.04 |
| ● | ○ | ○ | ○ | ● | ● | ● | ● | SAGE | **80.33 ± 2.98** | **81.11 ± 2.69** | **81.90 ± 1.91** | **80.33 ± 2.98** |
| | | | | | | | | GAT | 77.11 ± 3.30 | 77.89 ± 3.02 | 78.68 ± 2.25 | 77.11 ± 3.30 |
| | | | | | | | | GCN | 75.00 ± 3.54 | 75.78 ± 3.26 | 76.57 ± 2.49 | 75.00 ± 3.54 |
| ● | ○ | ○ | ● | ● | ○ | ● | ○ | SAGE | **89.27 ± 3.62** | **89.22 ± 3.66** | **91.64 ± 2.92** | **89.27 ± 3.62** |
| | | | | | | | | GAT | 86.05 ± 3.94 | 86.00 ± 3.99 | 88.42 ± 3.26 | 86.05 ± 3.94 |
| | | | | | | | | GCN | 83.94 ± 4.18 | 83.89 ± 4.23 | 86.31 ± 3.50 | 83.94 ± 4.18 |
| ● | ○ | ○ | ● | ● | ● | ● | ● | SAGE | **82.60 ± 2.20** | **81.09 ± 3.29** | **80.49 ± 4.42** | **82.60 ± 2.20** |
| | | | | | | | | GAT | 79.38 ± 2.52 | 77.87 ± 3.62 | 77.27 ± 4.76 | 79.38 ± 2.52 |
| | | | | | | | | GCN | 77.27 ± 2.76 | 75.76 ± 3.86 | 75.16 ± 5.00 | 77.27 ± 2.76 |
| ● | ● | ○ | ● | ● | ○ | ● | ○ | SAGE | **91.14 ± 4.79** | **90.73 ± 4.81** | **90.33 ± 5.03** | **91.14 ± 4.79** |
| | | | | | | | | GAT | 87.92 ± 5.11 | 87.51 ± 5.14 | 87.11 ± 5.37 | 87.92 ± 5.11 |
| | | | | | | | | GCN | 85.81 ± 5.35 | 85.40 ± 5.38 | 85.00 ± 5.61 | 85.81 ± 5.35 |
| ● | ● | ○ | ● | ● | ● | ● | ● | SAGE | **85.37 ± 2.11** | **84.90 ± 0.82** | **84.34 ± 4.44** | **85.37 ± 2.11** |
| | | | | | | | | GAT | 82.15 ± 2.43 | 81.68 ± 1.15 | 81.12 ± 4.78 | 82.15 ± 2.43 |
| | | | | | | | | GCN | 80.04 ± 2.67 | 79.57 ± 1.39 | 79.01 ± 5.02 | 80.04 ± 2.67 |
| ● | ● | ● | ○ | ● | ○ | ● | ○ | SAGE | **93.79 ± 1.80** | **93.79 ± 1.81** | **93.96 ± 2.05** | **93.79 ± 1.80** |
| | | | | | | | | GAT | 90.57 ± 2.12 | 90.57 ± 2.14 | 90.74 ± 2.39 | 90.57 ± 2.12 |
| | | | | | | | | GCN | 88.46 ± 2.36 | 88.46 ± 2.38 | 88.63 ± 2.63 | 88.46 ± 2.36 |
| ● | ● | ● | ○ | ● | ● | ● | ● | SAGE | **84.50 ± 1.60** | **81.37 ± 1.74** | **78.67 ± 1.90** | **84.50 ± 1.60** |
| | | | | | | | | GAT | 81.28 ± 1.92 | 78.15 ± 2.07 | 75.45 ± 2.24 | 81.28 ± 1.92 |
| | | | | | | | | GCN | 79.17 ± 2.16 | 76.04 ± 2.31 | 73.34 ± 2.48 | 79.17 ± 2.16 |
| ● | ● | ● | ● | ● | ○ | ● | ○ | SAGE | **98.87 ± 1.60** | **98.91 ± 1.59** | **98.95 ± 1.48** | **98.87 ± 1.60** |
| | | | | | | | | GAT | 95.65 ± 1.92 | 95.69 ± 1.92 | 95.73 ± 1.82 | 95.65 ± 1.92 |
| | | | | | | | | GCN | 93.54 ± 2.16 | 93.58 ± 2.16 | 93.62 ± 2.06 | 93.54 ± 2.16 |
| ● | ● | ● | ● | ● | ● | ● | ● | SAGE | **91.79 ± 3.62** | **91.70 ± 3.52** | **92.02 ± 3.52** | **91.79 ± 3.62** |
| | | | | | | | | GAT | 88.57 ± 3.94 | 88.48 ± 3.85 | 88.80 ± 3.86 | 88.57 ± 3.94 |
| | | | | | | | | GCN | 86.46 ± 4.18 | 86.37 ± 4.09 | 86.69 ± 4.10 | 86.46 ± 4.18 |

Table 9: Comparative analysis of different Graph Neural Network architectures (SAGE, GAT, GCN) on the GENRAD dataset. Results show performance across different data modalities, edge connections, and classification tasks. The filled circle indicates inclusion in the analysis, and the empty circle indicates exclusion.