# OpenReview forum: "GENRAD: Genomics and Radiomics Heterogeneous Graph Neural Network for Graph-Level Classification in Alzheimer's Disease"
_ICLR.cc/2025/Conference — Submitted to ICLR 2025_

### Official Review · Reviewer_24i2 · 2024-10-23

**Soundness:** 1
**Presentation:** 2
**Contribution:** 2
**Rating:** 5
**Confidence:** 4

**Summary:**

This paper proposes a heterogeneous GNN that integrates multimodal genomic and radiomic data for graph-level classification in Alzheimer’s Disease. The method represents patients, genes, and brain structures as distinct nodes and employs advanced message-passing techniques to improve classification performance.

**Strengths:**

1. The idea of incorporating neuroimaging data with demographic and genetic information is well-founded and addresses a relevant need in AD research.
2. The experimental results are impressive, showing strong performance on ANMerge dataset.

**Weaknesses:**

1. **Questionable Radiomic Feature Construction**: The choice of radiomic feature construction is unclear. Numerous established brain atlases, both anatomical [1] and functional [2], have been widely used in neuroscience and GNN research [3, 4] to construct brain networks. It is not explained why the authors opted for a segmentation model instead of utilizing these well-established atlases to generate structural data. Additionally, using 3D Euclidean distance to construct structure-structure edges seems questionable. Physical proximity between regions does not necessarily imply similar behavior or co-activation. The common practice in functional brain network construction typically involves using correlation metrics, not physical distance, to define edges.

2. **Limited Literature Discussion**: The paper lacks sufficient discussion on prior work that incorporates MRI and demographic data using GNNs. Existing works [5-8] in this area should be acknowledged and incorporated into both the discussion and experimental comparisons to contextualize the contribution of the proposed method.

3. **Missing Data Statistics**: Key data statistics, such as the number of patients, the distribution of classes, age and gender are missing. This information is crucial for transparency and reproducibility and should be included in the paper.

4. **Presentation Issues**: The presentation could be improved for better clarity:
   - (1) The figure on page 5 lacks a caption.
   - (2) In the first sentence of Section 3.2.3, "Figure 2b" should refer to "Figure 2a."
   - (3) The text and edges in Figure 3 are difficult to read and should be made clearer.
   - (4) I suggest revising Table 2 by separating binary classification results from multi-class classification results, as these settings are not directly comparable.

[1] Automated anatomical labeling of activations in spm using a macroscopic anatomical parcellation of the mni mri single-subject brain. Neuroimage 2002

[2] Local-global parcellation of the human cerebral cortex from intrinsic functional connectivity mri. Cerebral cortex 2018

[3] Data-driven network neuroscience: On data collection and benchmark. NeurIPS 2023

[4] NeuroGraph: Benchmarks for Graph Machine Learning in Brain Connectomics. NeurIPS 2023

[5] Self-attention equipped graph convolutions for disease prediction. ISBI 2019

[6] Multicenter and multichannel pooling GCN for early AD diagnosis based on dual-modality fused brain network. TMI 2022

[7] Multi-modal graph learning for disease prediction. TMI 2022

[8] DeepASD: a deep adversarial-regularized graph learning method for ASD diagnosis with multimodal data. Translational Psychiatry, 2024

**Questions:**

Please refer to weaknesses

---

> ### Author Response · Authors · 2024-11-19
> **Response to Reviewer  24i2**
>
> We are grateful for your insightful review. The points raised will be addressed comprehensively in our revisions, and the weaknesses highlighted will be addressed thoroughly in our response.
>
> W#1: Our dataset included only structural MRI, lacking fMRI or other neuroimaging modalities, which limited us to structural connections rather than functional correlations. Therefore, using physical proximity for structure-structure edges was a necessary approximation due to the absence of functional data, as established functional connectivity patterns in healthy brains may not translate to an AD cohort. After carrying out an ablation study with and without this edge attribute, as outlined in Table 2, the edge was able to boost the model’s performance.
>
> Moreover, we opted for a segmentation model specifically trained on a broad patient demographic that includes aging and diseased populations. This choice aligns well with our study focus on AD patients, as the segmentation model accounts for age-related and pathological variations in brain structure, which may not be represented in standard atlases typically constructed from healthy populations. Over the years, deep learning models have shown better and more robust results rather than using the atlas-based method. We have also validated the segmentation output with an expert radiologist for verification. We will clarify these methodological choices in the revised manuscript.
>
> W#2: We appreciate the suggestion to incorporate prior work integrating MRI and clinical data using GNNs. In the revised manuscript, we will expand our discussion to include relevant studies that have explored similar data modalities. By examining these works, we aim to better contextualize GENRAD's unique contributions, particularly in its approach to multimodal fusion and its focus on AD-specific pathology.
>
> W#3: Thank you for highlighting the need for detailed data statistics. Due to page limitations, we omitted these details in the main text since ANMerge is a public dataset, and this information can be found in the reference. Still, we agree that they are essential for transparency and reproducibility. We will add key data statistics—including the number of patients, class distribution, and age and gender demographics—to the appendix in the revised manuscript to ensure this information is readily accessible to readers.
>
> W#4: Thank you for the detailed feedback on presentation clarity. We appreciate these suggestions and will implement the mentioned changes to improve readability and organization.

---

> > ### Comment · Reviewer_24i2 · 2024-11-26
> >
> > The revision and response address some of my concerns. I have raised my score. However, the author’s justification for using 3D Euclidean distance to construct edges in brain networks is still not convincing. There exist numerous anatomical brain atlases that are widely used to define ROIs and generate brain networks. These atlases are grounded in neuroscientific evidence and offer a more biologically informed approach to edge construction.

---

### Official Review · Reviewer_RnJG · 2024-10-29

**Soundness:** 2
**Presentation:** 3
**Contribution:** 2
**Rating:** 3
**Confidence:** 5

**Summary:**

This paper introduces GENRAD, a novel heterogeneous graph neural network that improves Alzheimer's Disease classification by integrating genomic and radiomics data through a graph structure representing patients, genes, and brain regions as nodes. The model achieves 91.70% classification accuracy while providing interpretable insights about disease pathology through explainable AI techniques and unsupervised gene clustering, advancing both diagnostic capabilities and potential treatment strategies.

**Strengths:**

Building heterogeneous graph from radiomic and genomic information is novel, based on the reviewer's knowledge.

The paper is clear and easy to follow.

Thorough ablation study has been provided.

Potentially new findings can be found from the interpretations.

**Weaknesses:**

The comparisons in Table 1 are not convincing. For example, to demonstrate that GENRAD could outperform Zheng et al (2018) and Maddalena et al (2022), the baseline models should be retrained on the ANMerge. It is not fair to just put their results on ADNI in the table.

It is not clear how authors defined the edge between patient node and structure/gene node. And why there is no direct interaction between structure and gene node? This is important for AD for several reasons, like the amyloid and tau can be accumulated in different regions with different extents. Besides, the structure-structure edges were built based on 3D Euclidean distance between different regions. Is this better than other ways like covariance matrix from cortical thickness, or the correlation between node of brain regions, etc?

Why SAGEConv was chosen? There are many, many other message passing layers out there that are feasible, which should be compared with.

Neither code nor data is provided, make it hard to evaluate the reproducibility. It could be beneficial for the cummunity if they can be shared, though it is not necessary at this stage.

Minor: Missing index and caption for figure 2.

**Questions:**

Please refer to the weakness.

---

> ### Author Response · Authors · 2024-11-19
> **Response to Reviewer RnJG**
>
> Thank you for taking the time to review our work. We will carefully address the weaknesses and questions outlined in your feedback.
>
> W#1: Thank you for highlighting this point. As explained in the paper, no published work uses the ANMerge dataset, which limits our ability to retrain baseline models directly on ANMerge. Given this constraint, we included the results from Zheng et al. (2018) and Maddalena et al. (2022) as reference points from ADNI to provide a general performance context. However, we agree that this could be confusing, so we will split the tables and move the ADNI reference results to the appendix. This will make Table 1 focus on comparing different methods on ANMerge that we implemented ourselves to demonstrate the strengths of the proposed GENRAD model within this dataset. This approach allowed us to provide a meaningful performance comparison under consistent conditions, highlighting GENRAD's advantages on ANMerge. We just opted to add the ADNI result as the only previous work published on Maddalena et al. (2023) compares their result to the previous work on ADNI Maddalena et al. (2022) and others, so we added it for reference. Still, we agree this is not a fair comparison and will move it to the appendix.
>
> W#2: We defined edges between patient nodes and structure/gene nodes to capture each patient's unique genomic and structural associations without assuming generalized structure-gene interactions, which can vary widely in AD. By linking genes and brain structures indirectly through patient nodes, GENRAD can model patient-specific relationships, allowing it to learn complex gene-structure interactions individually. We used 3D Euclidean distance to define structure-structure edges due to the dataset's lack of functional or cortical thickness data. Although physical proximity is an approximation, it provides a workable structural framework. In Table 2, we have tested the influence of these structure edges, and they have been shown to improve the performance. In future work, with access to richer data, we aim to explore biologically informed alternatives like covariance from cortical thickness or functional correlations.
>
> W#3: As mentioned in the manuscript, we conducted experimentation with various message-passing layers to determine the most effective approach for GENRAD. Through these trials, SAGEConv consistently demonstrated the best performance for our specific task of integrating multimodal data for AD classification for multiple reasons, as pointed out in our response to Reviewer oYXN. The results using different message passing will be added in the appendix in the revised manuscript for reference. SAGEConv is well-suited for integrating genomic and radiomic data due to its scalable neighborhood sampling, which efficiently handles the sparse interactions in genomics and dense features in radiomics, and its flexible aggregation methods that adapt to the unique properties of each modality. Additionally, it reduces overfitting by learning localized representations while preserving meaningful cross-modal interactions during end-to-end training. In the revised manuscript, we will clarify this selection process to highlight the empirical basis for choosing SAGEConv.
>
> W#4: The dataset used in our study is publicly available and referenced in the manuscript. Additionally, we are committed to supporting reproducibility and plan to release the code for GENRAD in the near future, allowing the research community to evaluate and build upon our work.

---

### Official Review · Reviewer_oYXN · 2024-11-01

**Soundness:** 1
**Presentation:** 2
**Contribution:** 2
**Rating:** 3
**Confidence:** 5

**Summary:**

This paper introduces GENRAD, a heterogeneous graph neural network designed to integrate genomic and radiomic data for Alzheimer’s Disease (AD) classification. GENRAD is structured to handle multimodal data, representing patients, genes, and brain structures as distinct nodes and using advanced message-passing techniques. It claims four major contributions: enhancing multimodal data fusion, creating adaptive multi-scale graph representations, improving explainability, and enabling unsupervised clustering of genes to identify biologically relevant pathways. The model demonstrates superior accuracy compared to other methods in classifying AD.

**Strengths:**

1. The paper is well-structured and easy to understand.
2. The paper provides visualization results that highlight key genomic markers and brain regions associated with Alzheimer’s Disease (AD).

**Weaknesses:**

1. Limited Technical Contribution. To handle the multi-modality data integrating genomic and radiomic information, this paper employs a message-passing technique originally proposed in [1] on a heterogeneous graph that models patients, genes, and brain structures with both local and global interactions. This existing approach constitutes the paper’s major and only technical contribution, making the overall contribution marginal.

[1] Hamilton, W., Ying, Z. and Leskovec, J., 2017. Inductive representation learning on large graphs. Advances in neural information processing systems, 30.

2. Incomplete Literature Review. This paper lacks a comprehensive overview of existing multimodal data fusion methods, making it unclear how GENRAD specifically advances this field. The following papers are highly relevant to this study.

[2] Gaiteri, C., Ding, Y., French, B., Tseng, G. C. and Sibille, E., 2014. Beyond modules and hubs: the potential of gene coexpression networks for investigating molecular mechanisms of complex brain disorders. Genes, brain and behavior, 13(1), pp.13-24.

[3] Gaiteri, C., Mostafavi, S., Honey, C. J., De Jager, P. L. and Bennett, D. A., 2016. Genetic variants in Alzheimer disease—molecular and brain network approaches. Nature Reviews Neurology, 12(7), pp.413-427.

[4] Bodalal, Z., Trebeschi, S., Nguyen-Kim, T. D. L., Schats, W. and Beets-Tan, R., 2019. Radiogenomics: bridging imaging and genomics. Abdominal radiology, 44(6), pp.1960-1984.

[5] Li, S. and Zhou, B., 2022. A review of radiomics and genomics applications in cancers: the way towards precision medicine. Radiation Oncology, 17(1), p.217.

[6] Wang, M., Roussos, P., McKenzie, A., Zhou, X., Kajiwara, Y., Brennand, K. J. and Zhang, B., 2016. Integrative network analysis of nineteen brain regions identifies molecular signatures and networks underlying selective regional vulnerability to Alzheimer’s disease. Genome medicine, 8, 1-21.

[7] Singh, G., Manjila, S., Sakla, N., True, A., Wardeh, A. H., Beig, N. and Spektor, V., 2021. Radiomics and radiogenomics in gliomas: a contemporary update. British journal of cancer, 125(5), pp.641-657.

3. Lack of Justification for Methodological Choices. The rationale behind certain technical decisions—such as using the SAGEConv layer for message-passing—lacks detailed justification specific to the properties of the multimodal data. Furthermore, the reasons for employing the GeneMANIA method to obtain co-expression scores and for defining edges between brain structures based on 3D Euclidean distance remain unclear.

4. Insufficient Evidence and Contextual Interpretation. On page 7, it is stated that GENRAD incorporates explainable AI techniques, such as GNNExplainer, alongside biological interpretability frameworks to make predictions that are not only accurate but also clinically meaningful. However, GNNExplainer is neither discussed in the methodology section nor tailored to the specific properties of multimodal data, which leaves the AI techniques used insufficiently explained. Additionally, while visualizations of gene interactions and brain regions affected by Alzheimer’s Disease (AD) are provided, they lack biological context and supporting evidence, raising questions about the relevance of the identified biomarkers and brain regions and their alignment with established neuroscience findings.

5. Limited Dataset. The model evaluation uses a single dataset (ANMerge), limiting the generalizability of the results. Moreover, the dataset details, including size and data distribution, are not fully described.

6. Comparative Analysis with Broader Baselines. To validate GENRAD’s performance further, comparisons with a more comprehensive set of baselines, particularly from recent transformer-based approaches, would strengthen the findings.

7. Marginal Performance Improvement. The classification results in Table 1 show that the proposed GENRAD achieves only a slight improvement over the follow-up methods and, in some cases, performs even worse.

8. Unclear Presentation. Numerous abbreviations are used without providing their full terms, and Figure 2 lacks a caption, affecting overall clarity.

**Questions:**

1. Could the authors elaborate on why SAGEConv was chosen over other potential GNN architectures? How does it specifically benefit the integration of genomic and radiomic data?

2. Given the single dataset used, how do the authors view GENRAD’s applicability to other AD datasets, and are there plans for further validation with more diverse data sources?

3. While explainable AI techniques are included, how do the authors envision clinicians using these insights in real-world applications? Providing case studies or examples could be beneficial.

---

> ### Author Response · Authors · 2024-11-19
> **Response to Reviewer oYXN**
>
> We appreciate your detailed review and constructive comments. The concerns and questions highlighted will be addressed thoroughly in our response.
>
> W#1: We do not claim that implementing the SAGEConv layer is a technical contribution but a tool integrated within our graph structure. We have referenced Hamilton (2017) for this. GENRAD’s main contributions, outlined in the introduction and abstract, include: (1) enabling multimodal fusion of genomic and radiomic data, uncovering biologically meaningful insights; (2) introducing adaptive multi-scale graph representations to capture complex interactions across biological scales relevant to AD pathology; (3) incorporating explainable AI techniques to analyze genomic markers and brain regions; and (4) unsupervised clustering of genes, identifying biological pathways for personalized treatment strategies. These contributions highlight GENRAD’s impact on AD research.
>
> W#2: We will broaden the literature review to include more multimodal data fusion methods, including the suggested references.
>
> W#3 & Q#1: SAGEConv was chosen for its scalability and efficient handling of large graphs. Its adaptive aggregation methods (e.g., mean, max-pooling, LSTM) optimize genomic (sparse, sequential) and radiomic (dense, spatial) data. By projecting high-dimensional features into a shared low-dimensional space, it reduces noise while preserving essential information. It also encodes multimodal relationships, such as gene co-expression and structure-feature correlations, supporting efficient data fusion for accurate AD classification. Due to data limitations, GeneMANIA was selected for reliable gene-gene associations and 3D Euclidean distance approximated spatial relationships. Both edge types improved performance (Table 2). We will expand this discussion in the methodology section.
>
> W#4: We first introduced GNNExplainer in the literature review to provide background on this existing tool and clarify its context. Since we did not develop the GNNExplainer, we referenced it directly in Section 4 rather than detailing it in the methodology. We will clarify implementation details in the revised manuscript. Biological context discussions were placed in the appendix due to page limits, prioritizing technical results in the main text. These discussions, in Section A.4, will be expanded for clarity.
>
> W#5: Due to page limitations, we omitted some dataset details; since ANMerge is publicly accessible, all relevant information is available. However, we will add key dataset statistics, including size and data distribution, in the appendix to enhance transparency and reproducibility.
>
> W#6: We acknowledge this limitation and have highlighted it in the future work section of the conclusion. Since ANMerge is a relatively new dataset, there are currently no established baselines, which limits direct comparative analysis. However, we plan to address this in future work by including comparisons with a broader set of baselines.
>
> W#7: In Table 1, we present results from both ADNI and ANMerge datasets. We understand the potential confusion regarding this as pointed by Reviewer RnJG, so we will split the table and move the ADNI comparison to the appendix for clarity. A closer examination of ANMerge results demonstrates that GENRAD consistently outperforms other methods, reinforcing its ability to handle complex, multimodal data. For the first three classification tasks, GENRAD achieves an average performance increase of 2.29%, which is a significant improvement given the complexity of the dataset. For the AD vs. VaD task, while there is a minor 1% decrease in accuracy, the F1-score—a more robust metric for the class imbalance—improves by 6.71%, indicating a stronger predictive capability in distinguishing these classes. We believe these performance gains validate GENRAD's design and its utility in AD-related classifications.
>
> W#8: We will define all abbreviations on first use and add a caption to Figure 2 for clarity and readability, improving content accessibility.
>
> Q#2: GENRAD was tested on ANMerge only due to the scarcity of large-scale AD datasets with genetic data. These constraints made ANMerge a valuable choice. Future work will address missing modalities, enabling the use of datasets with partial information.
>
> Q#3: GENRAD’s explainable AI techniques aim to provide actionable insights for clinicians. Visualizations such as genomic markers, brain regions, and gene interaction diagrams bridge model outputs and clinical interpretation. Unsupervised gene clustering highlights related biological pathways, supporting personalized treatment strategies and making outputs accessible without requiring technical expertise. This was addressed in section A.4 in the appendix.

---

> > ### Comment · Reviewer_oYXN · 2024-11-22
> >
> > I appreciate the authors' detailed responses to my comments. My main concern still lies in the novelty and contributions of this work. While the authors have reiterated their contributions, the direct integration of existing techniques cannot be considered a significant contribution.
> >
> > Furthermore, the authors justify their choice of SAGEConv based on its scalability and ability to efficiently handle large graphs, along with experiments comparing various message-passing layers to achieve the best performance. This approach appears more aligned with an engineering application rather than addressing a specific scientific problem. It lacks consideration of the data's unique properties and does not clearly define the scientific challenges tackled.
> >
> > Adapting established methods to address the specific characteristics of newly fused multimodal data is recommended. However, this paper does not clearly highlight the unique properties of the data that distinguish it from other datasets.
> >
> > In summary, while the authors have made efforts to respond to feedback, the paper does not sufficiently demonstrate novelty or address distinct scientific research problems in its current state. I will retain my current score.

---

> > > ### Author Response · Authors · 2024-11-22
> > > **Response to Reviewer oYXN - Clarifying GENRAD's Contributions and Alignment with ICLR Scope**
> > >
> > > We thank the reviewer for their thoughtful feedback. We want to clarify that GENRAD's primary contributions align strongly with several core areas within ICLR 2025's scope, namely Learning on Graphs and Other Geometries & Topologies, Representation Learning for Multiple Modalities, Interpretability and Visualization, and Applications to Neuroscience & Biological Sciences. GENRAD uniquely lies at the intersection of these domains, combining graph-based methods, multimodal integration, and biological interpretability into a cohesive framework that addresses complex challenges in neuroscience and AI. Below, we address the main concerns raised, particularly regarding the novelty and alignment of our contributions with scientific challenges:
> > >
> > > We admit that rather than focusing on architectural changes; our primary contribution lies in advancing representation learning at the input for heterogeneous biological data through the novel graph-based integration of multimodal data and the model space regarding the hierarchical representation learning via supernodes and the interpretable multiscale biological insights. This contribution is central to ICLR's scope, especially since GENRAD presents a unique approach not only to address the scalability and heterogeneity of multimodal data but also to advance representation learning with direct translational value for Alzheimer's research, aligning strongly with ICLR's mission.
> > >
> > > GENRAD addresses core scientific challenges, including data heterogeneity, scalability of multimodal models, biological interpretability, and multiscale biological insights. The novelty of our work lies in the design of a heterogeneous graph that integrates multimodal data, combining biologically informed gene selection, co-expression relationships, and radiomic features extracted from segmented ROIs in brain MRI scans. Our multiscale graph captures both local and global interactions across modalities, providing a comprehensive representation of Alzheimer's Disease pathology. Additionally, we introduce supernodes for unsupervised clustering of genes based on shared co-expression patterns, improving interpretability and yielding insights into key pathways like APP and Tau while achieving impressive accuracy performance. Advanced explainability techniques further enhance the model by isolating critical subgraphs and biomarkers, directly linking predictions to actionable biological insights.
> > >
> > > We believe this work makes a meaningful contribution by addressing significant scientific challenges in AD diagnosis through an innovative graph-based framework. We hope this clarification addresses the reviewer's concerns and welcome further discussion.

---

### Official Review · Reviewer_JZWM · 2024-11-02

**Soundness:** 3
**Presentation:** 2
**Contribution:** 2
**Rating:** 3
**Confidence:** 5

**Summary:**

The paper introduces GENRAD, a heterogeneous graph neural network (GNN) designed for Alzheimer’s Disease (AD) classification through the integration of genomic and radiomic data. This model represents patients, genes, and brain structures as nodes, allowing complex interactions to be analyzed using advanced message-passing techniques.

**Strengths:**

1. The model’s use of multi-scale graph representations captures local and global biological interactions, providing a holistic understanding of AD pathology.
2. The unsupervised clustering of genes within the model facilitates the identification of functional biological pathways, which could aid in personalized treatment strategies.

**Weaknesses:**

1.  The paper lacks a thorough discussion of related interpretable GNNs specifically designed for AD. Many studies have developed explainable GNNs for AD, such as those in [1, 2, 3, 4]. A comparison of these models with GENRAD in terms of both design and interpretability effectiveness is necessary to justify the interpretability of the proposed model.

[1] Li, Xiaoxiao, et al. "Braingnn: Interpretable brain graph neural network for fmri analysis." Medical Image Analysis 74 (2021): 102233.
[2] Kim, Mansu, et al. "Interpretable temporal graph neural network for prognostic prediction of Alzheimer’s disease using longitudinal neuroimaging data." 2021 IEEE International Conference on Bioinformatics and Biomedicine (BIBM). IEEE, 2021.
[3] Zhou, Houliang, et al. "Interpretable graph convolutional network of multi-modality brain imaging for alzheimer’s disease diagnosis." 2022 IEEE 19th International Symposium on Biomedical Imaging (ISBI). IEEE, 2022.
[4] Xiao, Tingsong, et al. "Dual-graph learning convolutional networks for interpretable alzheimer’s disease diagnosis." International Conference on Medical Image Computing and Computer-Assisted Intervention. Cham: Springer Nature Switzerland, 2022.

2. While the paper claims computational efficiency, it does not provide theoretical or experimental evidence to substantiate this claim. A computational analysis section, including runtime comparisons, would justify the argument.

3. The paper does not provide a clear problem definition, which makes it hard to follow.

4. The mathematical equations contain typos and undefined symbols. For example, there is a mathematical symbol error in Equation (3) $\mathcal{N} _{\phi}\left( i \right)$ and undefined dimensions for variables.

5. From the Table 1, the performance of proposed methods is limited compared with other baselines, such as FGCNN, especially no standard deviation are provided.

**Questions:**

1. How does GENRAD compare in interpretability and model design with other interpretable GNNs?

2. Could the author analyse the interpretable experimental results in medical domain perspective?

3. Could the author provide a detailed computational efficiency analysis, both theoretical and experimental, to support the efficiency claims?

---

> ### Author Response · Authors · 2024-11-19
> **Response to Reviewer JZWM**
>
> Thank you for your review and valuable feedback. We will address the weaknesses and questions you raised below:
>
> W#1 & Q#1: We acknowledge the importance of situating GENRAD within the context of existing interpretable GNNs for AD. While our focus was primarily on introducing GENRAD's unique approach to multimodal data fusion and patient-specific insights, we agree that discussing prior interpretable GNN models in AD research will strengthen the paper. In the revised manuscript, we will include a comparative analysis with relevant studies, specifically addressing models' design and interpretability features like those cited.
>
> Additionally, GENRAD introduces multi-scale interpretability through its adaptive graph representations, capturing interactions at different biological scales essential for understanding AD pathology. Through GENRAD, we can extract gene-to-gene interaction, brain structure-to-structure interaction, and the multi-genic influence. These were highlighted in contributions numbers 3 and 4. Still, we will emphasize GENRAD's unique strengths in the revised manuscripts.
>
> Q#2: The interpretability of GENRAD is addressed in Section 4 and further addressed in the appendix, where we provide a detailed analysis of the unsupervised clustering of genes. This analysis examines how the identified gene clusters align with known AD-related biological pathways, such as those involved in amyloid processing and neuroinflammation. By linking these clusters to established AD mechanisms, we demonstrate the clinical relevance of GENRAD's outputs, offering insights that could support diagnostic and therapeutic strategies in the medical domain. These additions reinforce GENRAD's potential as a tool for clinically meaningful insights into AD pathology.
>
> W#2 & Q#3: While GENRAD was designed with efficiency in mind—using techniques like optimized message-passing and selective edge construction tailored to multimodal AD data—we agree that empirical validation is essential to substantiate this claim. In the revised manuscript, we will include a computational analysis section with model parameters and FLOPS details and comparisons against baseline models on equivalent hardware. This will provide evidence of GENRAD's efficiency advantages and demonstrate how its design uniquely balances computational demands with high performance on complex multimodal data.
>
> W#3: The problem GENRAD addresses—integrating complex multimodal data (genomic and radiomic) for Alzheimer's Disease (AD) classification while enhancing interpretability and capturing patient-specific interactions—is outlined in the paper, but we acknowledge that it may benefit from further emphasis. In the revised manuscript, we will make this problem definition more prominent in the introduction to ensure readers understand GENRAD's focus and its relevance to advancing Dementia research. This enhancement will make the paper's objectives easier to follow and highlight GENRAD's unique approach within the field.
>
> W#4: As defined, the notation Nϕ(i) represents the set of neighboring nodes of type ϕ of node i. We opted for a general, dynamic, and scalable formula that can accommodate a variable number of modalities without explicit dimension restrictions, which is why detailed dimensional information was not initially provided. In response to your feedback, we will carefully review all equations, include a notation for the number of modalities K, and include the dimension. This addition will enhance the mathematical clarity and precision of the manuscript.
>
> W#5: As noted in the paper, the lack of published results on the same dataset (ANMerge) limits direct performance comparisons with established baselines. Therefore, we implemented several ML and DL methods, including FGCNN, to set a standard for ANMerge performance and offer a consistent baseline for future studies. We focused on assessing GENRAD's performance independently within this context. Regarding the standard deviation, we did include these values in Table 2 for our method, as we recognize the importance of reporting standard deviations for a more comprehensive understanding of model robustness. Due to the limited space, the values for the other tested methods were omitted, but we will add them now since we will be splitting the table and will no longer need the dataset column.

---

> > ### Comment · Reviewer_JZWM · 2024-11-26
> >
> > Thank you for the response. While some concerns, such as W#1, are partially addressed, significant issues remain unresolved. For example, W#2 lacks evidence for computational efficiency claims, and W#3 requires a clearer problem definition. I will maintain my initial scores.

---

> > > ### Author Response · Authors · 2024-11-27
> > > **Response Regarding W#2 and #3**
> > >
> > > Thank you for your feedback. To address Weakness #2, we have included Table 8 in the updated manuscript, which provides a detailed comparison of GENRAD and a standard GCN model, demonstrating GENRAD’s computational efficiency in terms of FLOPs, inference time, and memory usage. For Weakness #3, we have refined the problem definition to underscore the paper’s focus on addressing the fundamental challenge of integrating and analyzing heterogeneous multimodal biological data. This work advances state-of-the-art predictive modeling while delivering biologically interpretable insights, specifically targeting complex diseases like Alzheimer’s Disease.
> > >
> > > We elaborate more below:
> > >
> > > **Addressing Weakness #2:**
> > > To validate the computational efficiency of GENRAD, we have added Table 8, which explicitly compares its performance against a standard GCN model. This table highlights the significant improvements GENRAD achieves in terms of FLOPs per layer, inference time, and memory usage, substantiating our claims with concrete metrics.
> > >
> > > **Addressing Weakness #3:**
> > > The central problem tackled by GENRAD is the effective integration and analysis of multimodal biological data to address challenges in:
> > > * Data Heterogeneity: Integrating modalities like genetics, radiomics, and clinical data while maintaining biological relevance.
> > > * Scalability: Handling large, complex datasets without compromising efficiency.
> > > * Biological Interpretability: Delivering actionable, multiscale insights into disease pathology.
> > > * Multiscale Representation Learning: Capturing local and global interactions critical to understanding disease mechanisms, such as APP and Tau pathways.
> > >
> > > GENRAD addresses these challenges using a heterogeneous graph design that integrates multimodal data, hierarchical representation learning through supernodes, and advanced explainability techniques. These features make GENRAD highly relevant to Alzheimer’s Disease research by bridging the gap between predictive modeling and biologically meaningful insights.
> > >
> > > **Additional Details in Our Response Document:**
> > > For further discussion of how GENRAD aligns with ICLR’s scope and its contributions to advancing representation learning at the input and model space, please refer to our detailed response titled "Response to Reviewer oYXN - Clarifying GENRAD's Contributions and Alignment with ICLR Scope." This document outlines GENRAD’s novel approach to integrating multimodal data and achieving scalable, interpretable representation learning with translational relevance to Alzheimer’s research.
> > >
> > > We hope these clarifications address your concerns and provide the necessary context for evaluating the manuscript.

---

### Author Response · Authors · 2024-11-22
**Manuscript Revision**

We would like to express our gratitude for the constructive feedback received from the reviewers, which has significantly improved the clarity and quality of our manuscript. Below, we outline the key revisions made and provide justifications for each change based on the reviewers' specific comments:

**Justification for Revisions**
**1. Splitting Table 1:** Based on feedback regarding the clarity of performance comparisons (e.g., W#7 from Reviewer RnJG), we moved the ADNI dataset details to the appendix to focus Table 1 exclusively on ANMerge results, avoiding confusion about baselines. As the old table created an unequal comparison that may not accurately reflect the strengths of GENRAD on ANMerge. By doing so, we aim to provide a fairer and more scientifically robust comparison, aligning the evaluation with the dataset-specific objectives of our study.

**2. Standard Deviation in Table 1:** Following the suggestion from Reviewer JZWM (W#5) to include standard deviation values for performance metrics, we added these to improve clarity and robustness in reporting.

**3. Dataset Class Split in Appendix:** To address concerns about missing dataset details (e.g., W#3 from Reviewer 24i2 and W#5 from Reviewer oYXN), we included class distribution and demographic statistics in the appendix for transparency and reproducibility.

**4. Message Passing Results in Appendix:** Reviewer oYXN (W#3) and Reviewer RnJG (W#3) questioned using SAGEConv. We added results from experimenting with alternative message-passing methods in the appendix, highlighting the empirical rationale for our choice.

**5. Computational Performance Results:** As suggested by Reviewer JZWM (W#2), we included computational performance analysis in the manuscript to substantiate claims of efficiency.

**6. Expanded Literature Review:** Reviewers JZWM (W#1), oYXN (W#2), and 24i2 (W#2) highlighted gaps in the literature discussion. We addressed this by incorporating comparative analyses with prior studies on interpretable GNNs and multimodal data integration.

**7. Edited Table 2:** Following Reviewer 24i2 (W#4), we separated binary and multi-class classification results for better comparability and readability. This change allows readers to better interpret the results within the appropriate task-specific context.

**8. Clarified Figure 2 Caption:** Reviewer RnJG and Reviewer 24i2 (W#4) noted ambiguities in Figure 2. We corrected the caption and ensured all abbreviations were defined for clarity.

**9. Updated Equation 3:** Reviewer JZWM (W#4) raised concerns about undefined dimensions and symbols. We revised Equation 3 to include a concrete example of modalities, improving mathematical clarity and contextual understanding.

We hope these updates comprehensively address the reviewers' feedback and enhance our work's overall quality and transparency. Thank you for considering our revised manuscript.

---

### Meta-Review · Area_Chair_hERy · 2024-12-16

**Metareview:**

This paper presents a method based on graph neural networks. The proposed model incorporates different data sources (here: genomics and radiomics data) to improve predictive performance on the ANMerge dataset. While the paper has its strengths, in particular the idea of combining different modalities (which in turn may also serve to improve explainability), the current manuscript suffers from several shortcomings, to wit:

- Lack of strong technical innovation: While the paper relies on established methods—and there is nothing wrong with that—an ICLR submission typical requires some additional insights into either the models, the representations, or the analyses. Beyond the predictive performance benefits on one dataset, no such insights are reported. Even the interpretability aspects, explicitly mentioned as a relevant feature of the method, relies on existing methods. Beyond the modelling decisions (that is, the graph creation), some of which have been criticised by reviewers (please see below), the task is interchangeable in the sense that [GNNExplainer](https://arxiv.org/abs/1903.03894) could be applied to other types of data and other tasks.

- Lack of additional comparison partners and datasets: Additional datasets would be helpful in assessing the claims of the paper; the reviewers also cited additional comparison partners that should be discussed (please see below). It should be noted that this is not the standard argument of reviewers and ACs giving authors a hard time by requiring more datasets. Rather, given the high relevance of the topic, an assessment of the generalisation performance (for instance) or the performance on a large-scale dataset would be highly useful. From the text, the well-established [ADNI dataset](https://adni.loni.usc.edu/) is also mentioned, but the training process on said dataset remains unclear.

- Lack of accessibility: The submission also suffers from accessibility and readability issues, mixing descriptions of a general model with specific implementations. There are numerous choices for the resulting model that are not justified (or ablated), for instance the selection of brain regions, which are [well-known to influence the results of a study](https://www.sciencedirect.com/science/article/pii/S1053811907011020).

It is for these reasons that a major revision, followed by another round of reviews, is required to fairly assess this paper. I therefore have to suggest rejecting this paper for now. I realise that this is not the desired outcome for the authors, so I hope that the discussion phase can be used to address some of the remaining issues for a resubmission. To add a suggestion to the authors: If this work were to be sent to a neurology venue and additional details/analyses were to be added, this has the potential to be a strong contribution to the literature. As it stands now, it will be hard to assess to what extent the performance improvements are suitable for any downstream applications. I understand that this is partially the fate of any paper attempting to bring ML into the applications, but I would strongly suggest the authors to either invest in a more detailed analyses of the results, or extend their paper with theoretical/model insights that pertain to AD.

**Additional Comments On Reviewer Discussion:**

Reviewers raised several highly-relevant points, including..

- ...a lack of contextualisation and discussion of related work (`JZWM`, `oYXN`, `24i2`)
- ...issues with feature construction and architectural choices (`oYXN`, `24i2`, `RnJG`)
- ...unfair comparisons or missing baselines (`oYXN`, `RnJG`)

The authors partially alleviated some of these points, for instance by adding preliminary citations to other works. This, and some clarifications, have been positively received by reviewers. However, during the rebuttal phase, the methodological concerns were _not_ alleviated. For instance, reviewer `oYXN`, after the rebuttal, raises the following point (highlighting added):

> Furthermore, the authors justify their choice of SAGEConv based on its scalability and ability to efficiently handle large graphs, along with experiments comparing various message-passing layers to achieve the best performance. This approach appears more aligned with an engineering application rather than addressing a specific scientific problem. **It lacks consideration of the data's unique properties and does not clearly define the scientific challenges tackled.**

This is echoed to a certain extent in the other interactions with the reviewers. While I agree with the authors that this paper is well in-scope for ICLR, it nevertheless would require substantial changes, in particular a stronger experimental setup _or_ a task-specific architecture (or a better analysis of its constituent components). As it stands, I thus agree with the assessment of the reviewers, giving the most weight to the methodological concerns, followed by the issues raised with the experimental setup.

I found the discussion phase to be very productive overall, and sincerely hope that the authors can use the feedback to improve their work.

---

### Decision · Program_Chairs · 2025-01-22

Reject